# Exercise improves the outcome of anticancer treatment with ultrasound-hyperthermia-enhanced nanochemotherapy and autophagy inhibitor

Chi-Feng Chiang[1]☯, Zi-Zong Wang[1]☯, Yu-Hone Hsu[2], Shi-Chuen Miaw[3]*, Win-Li Lin[1,4]*

1 Department of Biomedical Engineering, National Taiwan University, Taipei, Taiwan, 2 Division of Neurosurgery, Department of Surgery, Kaohsiung Veterans General Hospital, Kaohsiung, Taiwan, 3 Graduate Institute of Immunology, College of Medicine, National Taiwan University, Taipei, Taiwan, 4 Institute of Biomedical Engineering and Nanomedicine, National Health Research Institutes, Zhunan Town, Miaoli, Taiwan

☯ These authors contributed equally to this work.
* winli@ntu.edu.tw (WLL); smiaw@ntu.edu.tw (SCM)

**Data Availability Statement:** All relevant data are within the paper and its Supporting information files.

## Abstract

It has been shown that exercise has a direct impact on tumor growth along with functional improvement. Previous studies have shown that exercise decreases the risk of cancer recurrence across various types of cancer. It was indicated that exercise stimulates the immune system to fight cancer. Previous study demonstrated that pulsed-wave ultrasound hyperthermia (pUH) combined with PEGylated liposomal doxorubicin (PLD) and chloroquine (CQ) inhibits 4T1 tumors growth and delays their recurrence. In this study, we investigated if the combinatorial treatment with high-intensity interval training (HIIT) combined with pUH-enhanced PLD delivery and CQ improved the outcome. The mouse experiment composed of three groups, HIIT+PLD+pUH+CQ group, PLD+pUH+CQ group, and the control group. HIIT+PLD+pUH+CQ group received 6 weeks of HIIT (15 min per day, 5 days per week) before 4T1 tumor implantation. Seven days later, they received therapy with PLD (10 mg/kg) + pUH (3 MHz, 50% duty cycle, 0.65 W/cm$^2$, 15min) + CQ (50 mg/kg daily). Results showed that HIIT+PLD+pUH+CQ significantly reduced the tumor volumes and brought about longer survival of tumor-bearing mice than PLD+pUH+CQ did. Blood cell components were analyzed and showed that neutrophil and reticulocytes decreased while lymphocytes increased after exercise.

## Introduction

Cancer remains the major cause of death in most parts of the world. In clinical settings, cancer is often depicted as tumors with uncontrolled cell growth and abnormal differentiation. The concept of "hallmarks of cancer", first proposed by D. Hanahan and R.A. Weinberg in 2000 and then updated in 2011 and 2022, described the complex and bizarre nature possessed by most types of cancer [1]. These hallmarks, including the capability to metastasize and the

**Funding:** This work was supported by the research grants from the Ministry of Science and Technology of Taiwan (MOST 109-2221-E-002-046).

ability to evade the immune system, distinguish cancer from normal biological tissue and make them threatening to human health.

Beyond these alterations happening in cellular or tissue level, cancer also causes systemic changes to the human body. Many cancer patients experience cancer-related fatigue, a common symptom that can have negative impacts on their physiological, psychological, and social functioning [2–7]. Cancer-related fatigue may be induced by the disease itself or by anticancer treatment. During chemotherapy, many patients experience severe weakness that may disturb their daily life [8]. It is suggested that exercise can relieve chemotherapeutic-related symptoms and improve the quality of life in cancer patients [9–11]. According to the guideline published by American College of Sports Medicine, exercise in cancer patients is feasible and is recommended because the benefit of exercise outweighs the harm for most patients [12]. The guidelines recommend engaging in moderate-intensity aerobic exercise for 150 minutes per week or vigorous-intensity aerobic exercise for 75 minutes per week. In addition, they recommend moderate-intensity muscle-strengthening activities for each major muscle group at least 2 days per week [12]. It is indicated that high-intensity exercise improves the general condition of cancer patients, such as cardiovascular function and muscle power, and alleviates cancer-related discomforts [13–19].

Besides functional improvement, there are a growing number of research studies indicating that exercise may also directly affect cancer cells. Several observational studies revealed that physical activity effectively reduced the recurrence of several types of cancer, including colorectal cancer, breast cancer, and prostate cancer [20–22]. In addition to epidemiological evidence, physical activity was also considered to influence multiple hallmark features of cancer that provided mechanistic explanations about the effects of exercise against cancer [23, 24]. Some studies have shown that exercise can enhance the translocation of lymphocytes into the bloodstream, which may activate the immune system to combat cancer [25]. It was found that natural killer (NK) cells were substantially recruited through the upregulation of catecholamines during exercise [26–28]. Apart from catecholamines, several cytokines including interleukin-15, interleukin-7, and interleukin-6 were reported to be involved in the exercise-related recruitment of NK cells [29]. It was also reported that exercise-induced IL-6 inhibited the proliferation of malignant cells through regulation of DNA damage [30]. Other factors such as blood flow, oxygen consumption, and body temperature may also play a role in the regulation of immune cells during exercise [31–35].

Hyperthermia has been used in the treatment of cancer for a long time. When the temperature is raised above a certain threshold for a certain amount of time, hyperthermia can cause protein denaturation, impair the cellular function in heated cells, and eventually lead to cell death [36–38]. In addition, hyperthermia stimulates the immune system against cancer by exposing tumor-derived neoantigens released during hyperthermia therapy [39–41]. Hyperthermia as an adjunctive therapy with standard anticancer treatment such as chemotherapy or radiotherapy effectively improved the response rate and survival [42, 43]. By increasing tissue perfusion and vascular permeability, hyperthermia effectively facilitates the delivery of chemotherapeutic compounds into tumors and strengthens their activity against malignant cells [44]. There are several thermal energy sources used in clinical hyperthermia, and ultrasound hyperthermia is advantageous due to its non-invasive nature, lack of irradiation, and deep penetration into the body.

Autophagy is an important physiological mechanism that allows cells to recycle cellular components and generates energy [45, 46]. Cancer cells rely on autophagy as an escape mechanism to deal with environmental stress [45–50]. Blocking autophagy has been proposed as a promising strategy to complement standard anticancer treatment [49–51]. Chloroquine (CQ), an antimalarial drug with a long history, has been repurposed as an autophagy inhibitor [49].

In addition to its well-known activity in inhibiting autophagy, CQ is also reported to possess several other properties that may be beneficial in the treatment of cancer [52, 53].

Previous study showed that short-time pulsed-wave ultrasound hyperthermia (pUH) enhanced the delivery and efficacy of PEGylated liposomal doxorubicin (PLD) against 4T1 murine breast cancer [54, 55]. Furthermore, with the assistance of chloroquine (CQ), the combinatorial therapy could achieve long-term suppression on tumor growth and delay the recurrence [56].

The aim of present study is to utilize the multi-faceted effect of exercise prior to hyperthermia-facilitated nano-chemotherapy and autophagy inhibitor, in hope of a comprehensive and effective treatment strategy. We subjected mice to high-intensity interval training (HIIT) before implanting 4T1 murine breast cancer cells, and subsequently treated the resulting subcutaneous tumors with a combinatorial strategy involving PLD, pUH, and CQ. Our objective was to investigate whether prior high-intensity exercise inhibited tumor growth and prolonged the survival of tumor-bearing mice.

## Materials and methods

### In vivo anti-tumor efficacy study

All in vivo animal experiments were approved by the Institution of Animal Care and Use Committee, College of Medicine, National Taiwan University. All personnel who carried out in vivo animal experiments were sufficiently trained and qualified by the Institution of Animal Care and Use Committee, College of Medicine, National Taiwan University. The study protocol was not prospectively registered.

The study consisted of three groups, the HIIT+PLD+pUH+CQ group (n = 6), the PLD +pUH+CQ group (n = 8), and the control group (n = 7). The number of animals was determined by the capacity of experiment instruments and operators. The schedule of treatment experiments was demonstrated as Fig 1.

For the HIIT+PLD+pUH+CQ group, mice received 6 weeks of HIIT (15 minutes per day, 5 days per week) and were thereafter implanted with 4T1-luc2 cells. Seven days after tumor implantation, they were intravenously injected with 10 mg/kg PLD via tail vein and were then subjected to pUH treatment. After PLD and pUH treatment, they were daily fed with 50mg/kg CQ (dissolved in water) via oral gavage until the end of experiment. The duration of whole experiment was 60 days. For the PLD+pUH+CQ group, the schedule was similar to that in the HIIT+PLD+pUH+CQ group except that they did not receive HIIT before tumor implantation.

### Outcomes of anti-tumor efficacy study

The principal outcome indicators for anti-tumor efficacy were the tumor growth response over time and the survival of tumor-bearing mice. The volumes of subcutaneous tumors were measured with digital caliper every three to four days and their volumes were calculated as 0.5*length*width*width. The time of recurrence was defined when the tumor growth curve has passed treatment-induced nadir and the tumor regrow to an extent exceeding its pre-nadir peak volume.

Survival was determined by disease-related death, tumor volume exceeding the pre-defined tumor burden limit, or moribundity. In accordance with animal welfare guidelines, when the volume of subcutaneous tumor exceeded 1000 mm$^3$, the tumor-bearing mouse was considered reaching humane endpoint and was euthanized immediately after tumor volume measurement by cervical dislocation under isoflurane anesthesia. In addition to tumor volume, the body weight and health condition of mice were observed every day for clinical signs indicating

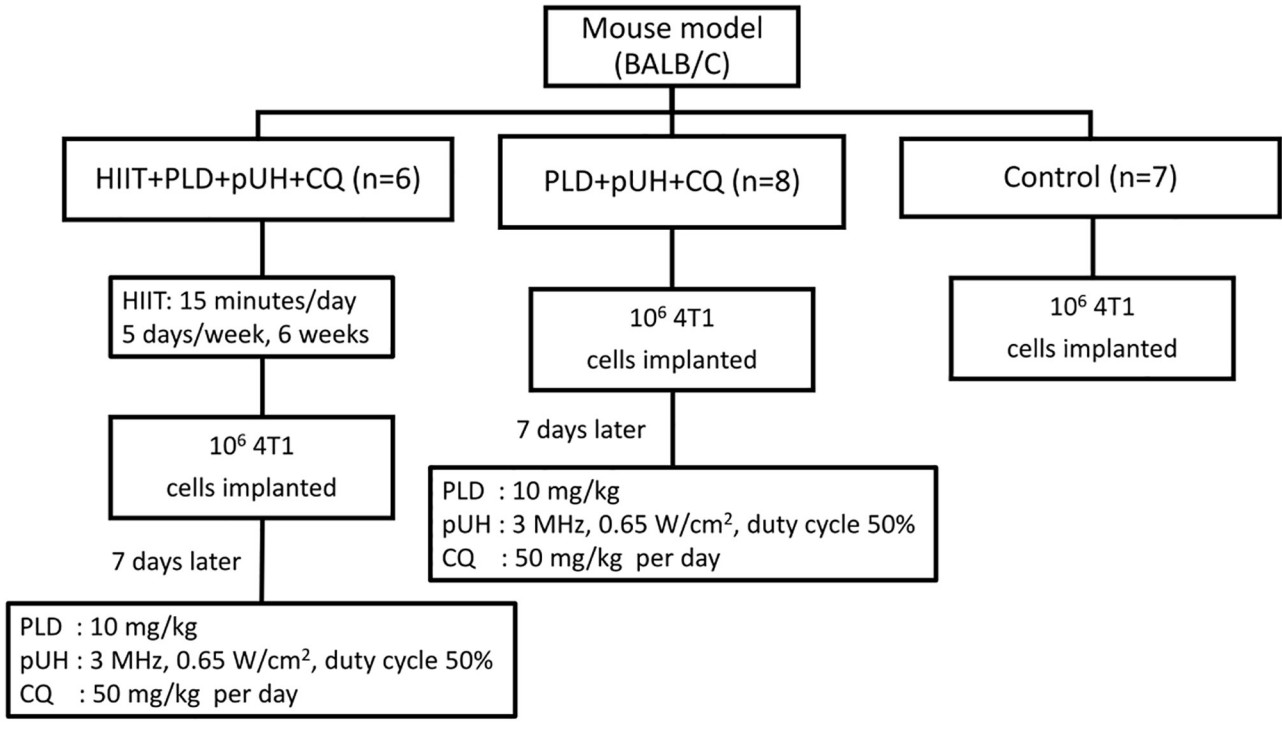

**Fig 1. The experimental design of in vivo animal study.**

severe discomfort due to growing cancer. In case of mice expressed signs for moribundity such as rapid decline in body weight (>20%), decreased appetite, or hunched posture, the mice were euthanized to end their suffering. The implanted tumors were also closely monitored for signs such as ulceration or bleeding. In case mice had tumors with unhealed ulcers or active bleeding and showed signs of severe discomfort, they were considered as reaching early end-point and were euthanized for the sake of humane consideration. No mice died before meeting criteria of euthanasia.

### Tumor cells and in vivo tumor model

In this study, 4T1 murine breast cancer cells (ATCC® CRL-2539TM) transferred with firefly luciferase 2 (4T1-luc2) and female BALB/c mice were used as the in vivo experimental model to demonstrate the efficacy of different treatments. 4T1-luc2 cells were cultured in Advanced Dulbecco's Modified Eagle Medium (DMEM) supplemented with 10% heat-inactivated fetal bovine serum (FBS), 1% Penicillin- streptomycin, and 1% glutamine in 10 cm tissue culture dishes in a 5% $CO_2$-containing incubator under 37˚C.

Female BALB/c mice were acquired from BioLASCO Taiwan Co., Ltd. and were housed within an environment with constant temperature and humidity and a 12-h light/dark cycle and were provided free access to water, standard diet, and environment enrichment. Experiments began when the mice reached the age of six weeks after one week of adaptation to avoid the effect of stress due to change in environment.

The in vivo tumor-bearing mice models were established by implanting 4T1-luc2 cells subcutaneously. Female BALB/c mice were anesthetized by 1–3% isoflurane inhalation during the tumor implantation procedure, and their hair on the right flank was removed before tumor implantation. Total $10^6$ of 4T1-luc2 tumor cells suspended in 100 μL of phosphate buffered

saline (PBS) were slowly injected subcutaneously into the right flank. Seven days after implantation while the volume of the subcutaneous tumor reached about 50~100 mm$^3$, the tumor-bearing mice were subjected to different treatment as specified in the above section.

## Chemical agents

The anticancer drug used in this study, PEGylated liposomal doxorubicin (PLD), was purchased from Taiwan Tung Yang Biopharm Co. Ltd. Chloroquine diphosphate was purchased from Sigma-Aldrich.

## Exercise treatment

The exercise treatment in this study was conducted by letting mice run in a wheel. The wheel had a diameter of 16 cm and was manually driven with a rate of one round every three seconds, equal to 16.75 cm/s running speed. Based on the relationship between running speed and maximal oxygen consumption (VO2max) in mice proposed by Fernand et al. [57], such running speed corresponds to 74.2% VO2max. It is equivalent to 82.5% HRmax according to the equation developed by Lounana et al. [58], exceeding the threshold of high intensity training. The high-intensity interval training (HIIT) exercise regimen in this study was conducted by letting mice run for 3 minutes, rest for 1.5 minutes, then repeated for 5 times per day. The training persisted for 6 weeks, with 5 days of exercise per week.

## Ultrasound settings and pulsed-wave ultrasound hyperthermia

The pulsed-wave ultrasound hyperthermia treatment in this study was conducted using a physical therapeutic system (US-750, ITO Physiotherapy & Rehabilitation). Before ultrasound hyperthermia, the mice were fixed on a platform under anesthesia with their subcutaneous tumors exposed upright. The ultrasound transducer was placed in a home-made water bag and immersed with degassed water. The probe and water bag was then mounted on top of the tumor to be treated, and the gap between water bag and skin was filled with acoustic gel for better ultrasound transmission. The ultrasound transducer circularly scanned above the tumor to spread thermal energy more uniformly. The parameters of ultrasound settings used in this experiment were frequency of 3 MHz, acoustic intensity of 0.65W/cm$^2$, pulse repetition frequency of 100 Hz 50% duty cycle, and duration of 15 minutes.

## Blood cell components analysis

Blood cell components pre-exercise and post-exercise were analyzed to understand the change in physiological environment induced by exercise. Four mice began six weeks of HIIT when they were 7-week-old. The details of HIIT were described in Exercise treatment section. The blood cell components data was collected before they started HIIT as a baseline (7-week-old) and was collected again after they completed six-week training course (13-week-old). Another six mice of the same age as the four mice receiving HIIT were used to control the effect of aging. They received no specific treatment, and the blood cell components were collected when they were 7-week-old and 13-week-old, respectively.

## Statistical analysis

The numerical values of experiments were expressed in the form of mean ± SEM (standard error of mean). The tumor growth curves were modelled with linear regression on log-transformed tumor volumes, and then the regression coefficients were analyzed. The survival of mice was analyzed with Log Rank (Mantel-Cox) test. Statistical significance was defined as

p<0.05. Analysis of tumor growth curves were performed with StatsNotebook, an open-source statistical package built on R. Survival analysis was performed with IBM SPSS Statistics version 20.0 (IBM Corp., Armonk, NY, USA).

## Results

### Exercise improved the anti-tumor efficacy of pulsed-wave ultrasound hyperthermia in combination with liposomal doxorubicin and chloroquine treatment in in vivo 4T1 tumor model

To investigate if exercise combined with PLD+pUH+CQ successfully inhibited tumor growth and prevented recurrence more effectively, we conducted in vivo experiment with 4T1-luc2 murine breast cancer model to see if exercise altered the growth of tumor and the response to PLD+pUH+CQ treatment. Mice received high-intensity interval training (HIIT) with wheel running for 15 minutes per day, 5 days per week, for a total of 6 weeks. After 6 weeks of HIIT, they were implanted with 4T1-luc2 murine breast cancer cells subcutaneously in their right flanks. Seven days after tumor implantation, tumor-bearing mice were subjected to PLD+pUH treatment and began daily intake of CQ via oral gavage, and the tumor volumes were closely monitored.

Fig 2(a) showed the log-transformed growth curves of tumors under different treatment condition, and Fig 2(b) showed the fitted line with linear regression model. It demonstrated that mice experiencing 6 weeks of HIIT prior to tumor implantation and treatment had smaller subcutaneous 4T1-luc2 tumors when compared to the mice treated with PLD+pUH+CQ but without exercise. The analysis of linear regression model was summarized in Table 1. The F-statistics and R-squared values showed that the model provided valid and good linear fit to the log-transformed tumor volumes. The p value corresponding to the explanatory variable of treatment was 0.00888, indicating that there was significant difference in tumor growth curves between HIIT+PLD+pUH+CQ group and PLD+pUH+CQ group. These findings suggested that the addition of exercise into combination treatment strengthened the anti-tumor efficacy of PLD+pUH+CQ treatment.

### Survival analysis

The survival of tumor-bearing mice was analyzed to evaluate the benefit of exercise, and the results were presented as Kaplan-Meier plot in Fig 3. Without treatment, tumor-bearing mice in the control group all died before Day 35 after tumor implantation (median survival 32 days). Treatment with PLD+pUH+CQ significantly prolonged the survival (median survival 42 days, p = 0.0013 as compared to the control group), and treatment with HIIT+PLD+pUH+CQ further lengthened the survival (median survival 56 days, p = 0.033 as compared to the PLD+pUH+CQ group, p = 0.00069 as compared to the control group).

### Body weight changes due to treatments

To see if the treatments caused any side effects to mice, the body weights of the mice were regularly measured every 7 days since D7 after tumor implantation. The changes in the body weights were shown in Fig 4. There was no substantial drop in body weight seen during the 60-day observation period. It implied that both treatments, HIIT+PLD+pUH+CQ and PLD+pUH+CQ, were well tolerated without obvious side effects.

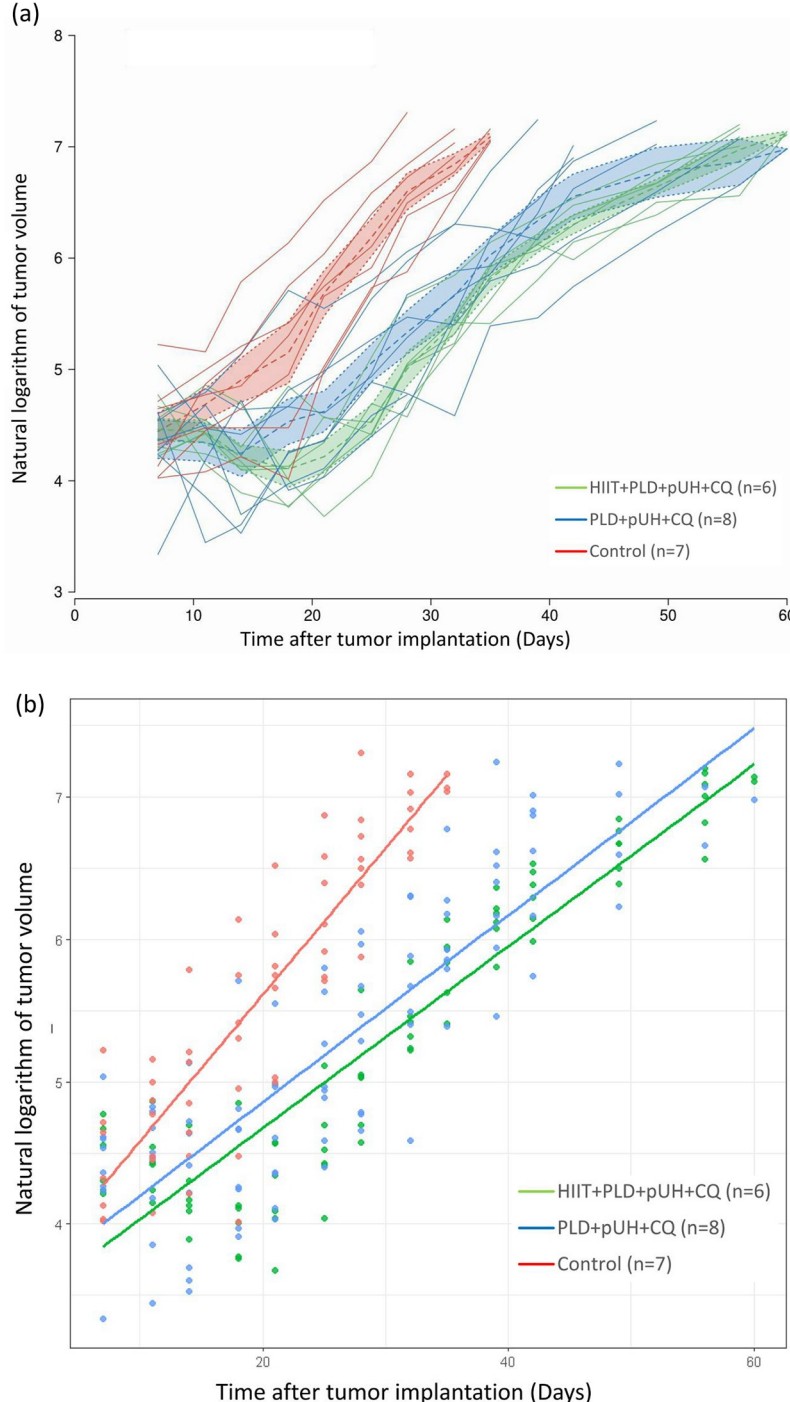

**Fig 2. Tumor growth trajectories and the response to different treatments.** (a) Log-transformed tumor growth curves of subcutaneous 4T1 murine breast cancer in response to different treatment: HIIT+PLD+pUH+CQ (green), PLD+pUH+CQ (blue), and control groups (red). Solid line represented individual growth trajectory of each tumor. Dashed line represented the mean tumor volume within group, while shaded area represented the standard error of mean. (b) Log-transformed tumor volumes represented as scatter plot with fitted line by linear regression model.

**Table 1. Summary of linear regression model of log-transformed tumor volumes.**

| Linear regression | | | | |
|---|---|---|---|---|
| HIIT+PLD+pUH+CQ group versus PLD+pUH+CQ group | | | | |
| Explanatory variables | Coefficient | Standard error | *t* statistics | *p* value |
| Time | 0.064681 | 0.002634 | 24.559 | $< 2*10^{-16}$ *** |
| Treatment | -0.196684 | 0.074287 | -2.648 | 0.00888 ** |

Note.

*** $p<0.001$,

** $p<0.01$,

* $p<0.05$

Multiple R-squared: 0.7832, Adjusted R-squared: 0.7806.

F-statistic: 301.7 on 2 and 167 DF, *p*-value: $< 2.2*10^{-16}$.

## Analysis of change in blood cells after HIIT

To further elucidate how exercise changed the physiological environment within mice and how these changes contributed to the treatment efficacy observed above, we analyzed the blood cell components before and after exercise in mice with HIIT (n = 4). The results were

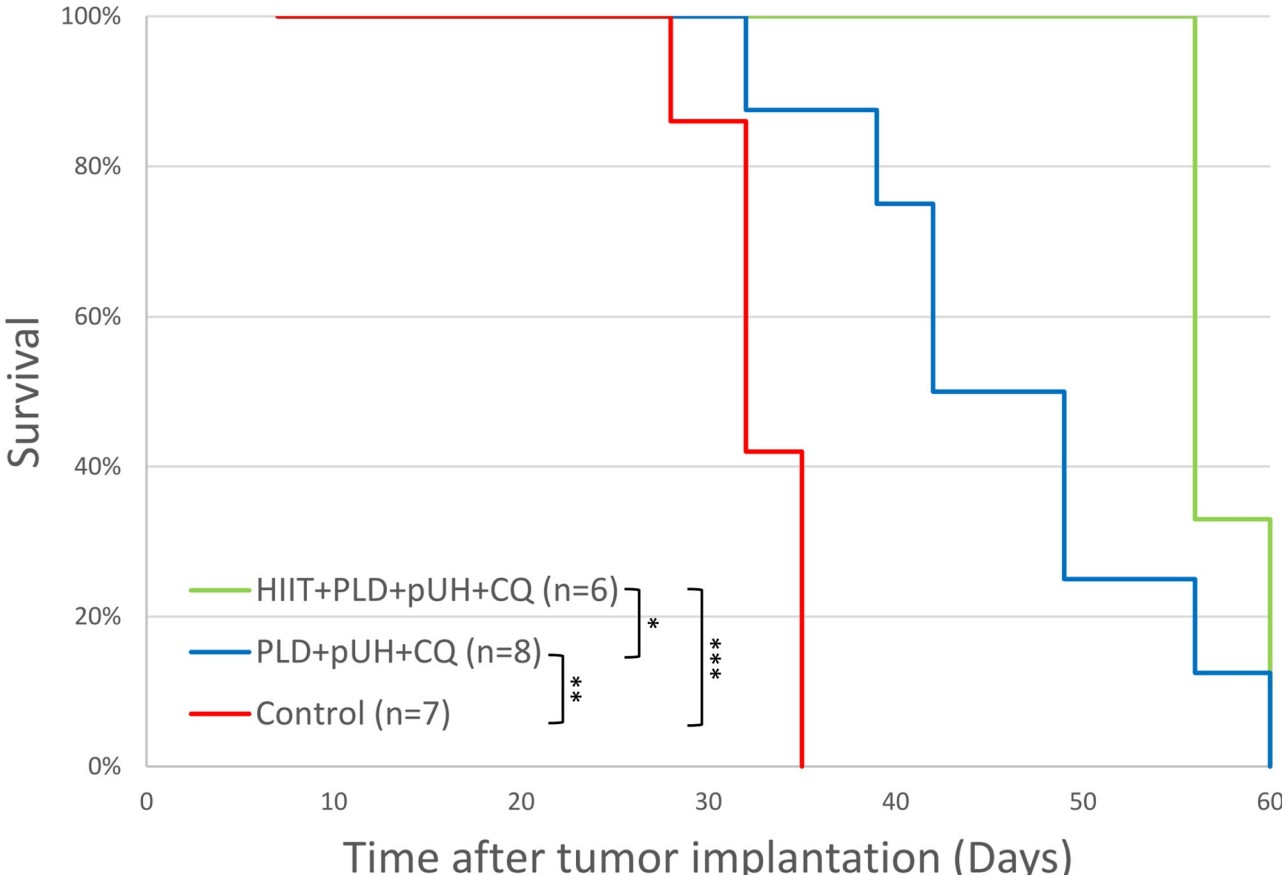

**Fig 3. Survival of tumor-bearing mice in response to treatments.** Kaplan-Meier survival plot for HIIT+PLD+pUH+CQ, PLD+pUH+CQ, and control groups. * p<0.05, ** p<0.01, *** p<0.001.

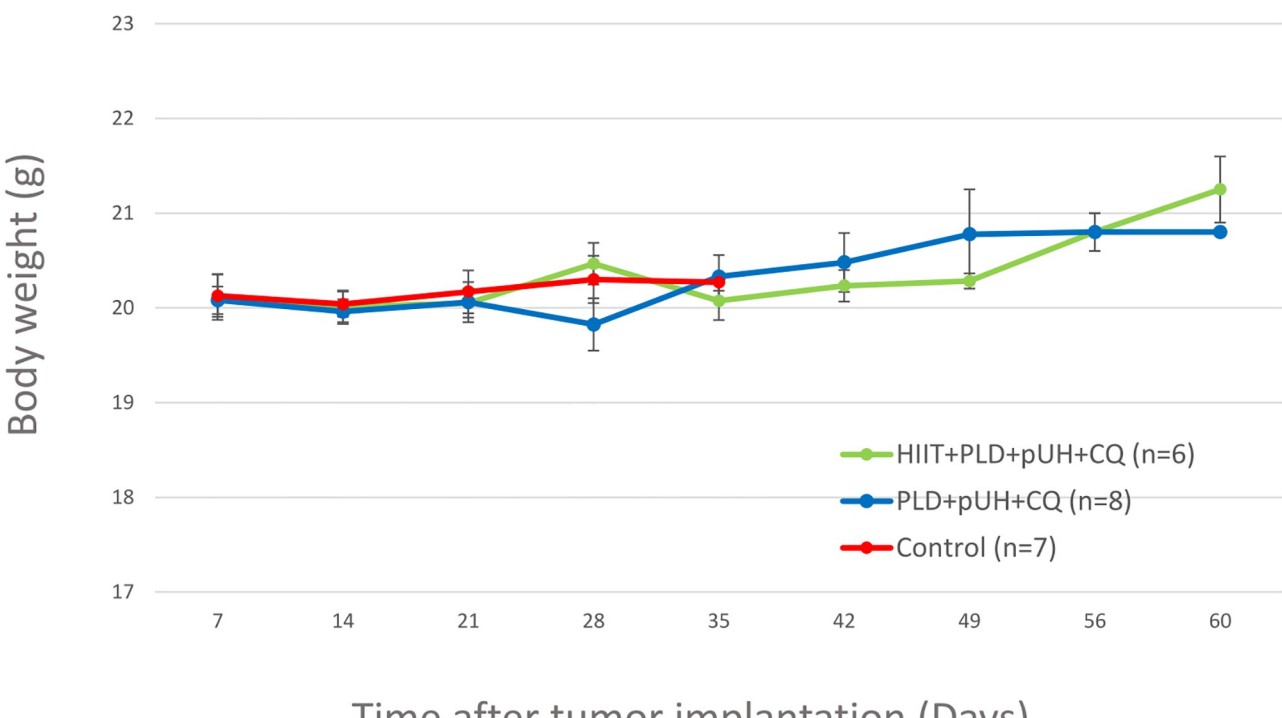

**Fig 4. Body weight changes during the observation period.**

compared to their baseline value which was derived from blood samples obtained before their 6-week training course. Besides, these values were also compared to those of mice which did not receive HIIT.

Fig 5 showed that the percentage of neutrophils among white blood cells (NEUT%) was significantly reduced in mice experiencing 6 weeks of HIIT, whereas in mice without exercise the change in NEUT% was not significant. On the other hand, the percentage of lymphocytes (LYMPH%) significantly increased after HIIT as demonstrated in Fig 6. As in the effect of exercise to oxygen-carrying red blood cells (RBC), the number of reticulocytes (RET) and their ratio among all RBC (RET%) were shown in Fig 7(a) and 7(b), respectively. The number and ratio of reticulocytes were both substantially reduced after 6 weeks of high intensity exercise.

## Discussion

In this study, we proposed a combinatorial treatment strategy by utilizing the multi-faceted effect of exercise to facilitate the efficacy of anticancer therapy. We incorporated high-intensity exercise with ultrasound hyperthermia and pharmaceutical agents, including nano-sized chemotherapeutic PLD and autophagy inhibitor CQ, to see if these components working in synergy to exhibit stronger efficacy against 4T1 tumor than PLD+pUH+CQ therapy did. We found that the tumor volume was significantly reduced by HIIT+PLD+pUH+CQ treatment than those treated with PLD+pUH+CQ. The tumor growth was markedly hindered by HIIT +PLD+pUH+CQ treatment, even causing slight reduction in tumor volume from Day 7 to Day 18. The tumors regained their ability to rapidly grow after Day 25. As a comparison, in the PLD+pUH+CQ group the duration of suppressed growth was shorter, and the time when

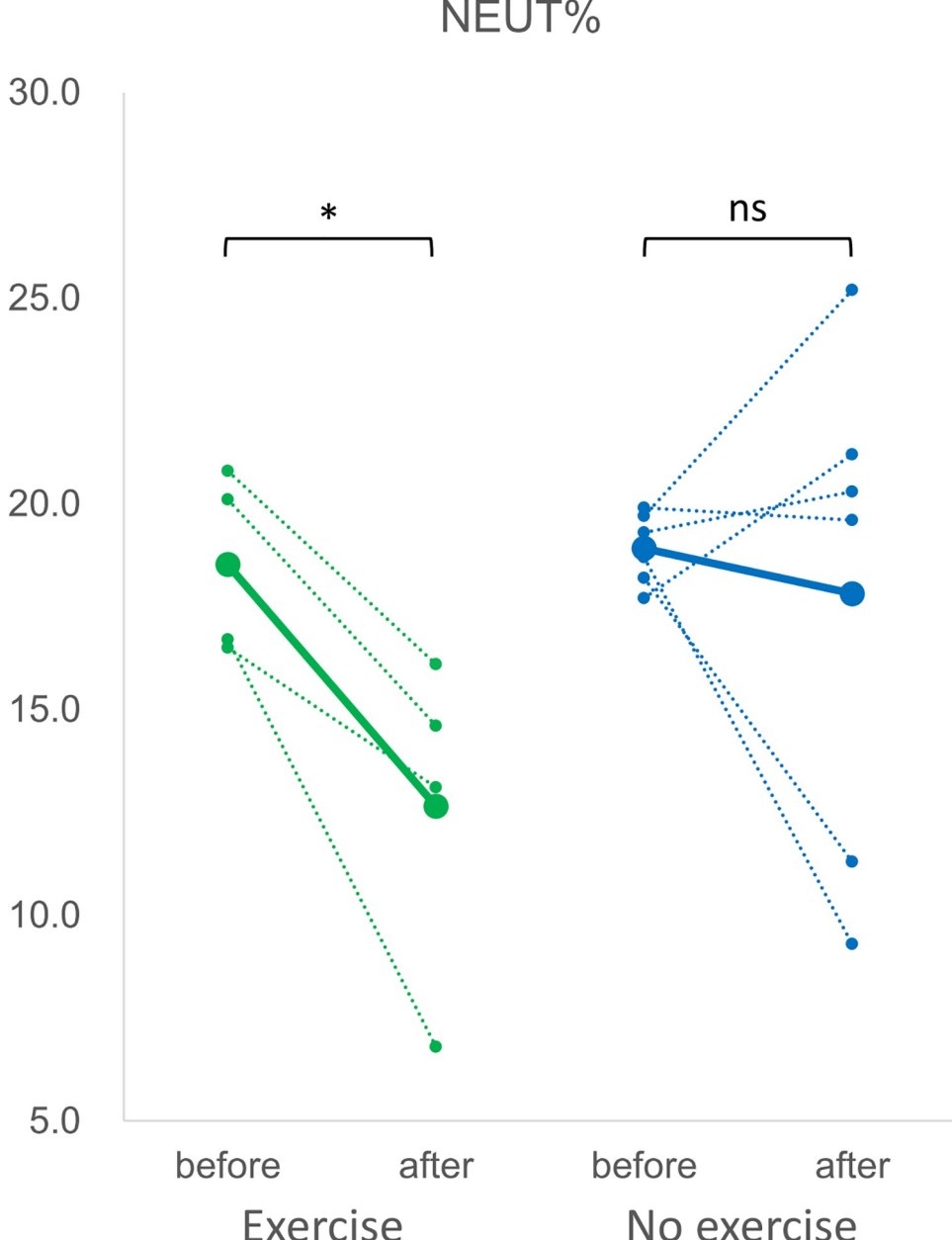

**Fig 5. The percentage of neutrophils among white blood cells (NEUT%).** The green dotted lines indicated the changes in NEUT% of individual mice in the exercise group obtained before and after 6 weeks of high-intensity exercise, and the green solid line indicated their mean value. The blue dotted lines were the changes in NEUT% of individual mice in the no exercise group before and after the duration of 6 weeks, and the blue solid line indicated their mean value. *: p<0.05. ns: not significant.

tumors start to regrow was also earlier. Furthermore, the survival time of tumor-bearing mice in the HIIT+PLD+pUH+CQ group was significantly longer than that in the PLD+pUH+CQ group. Exercise may improve the efficacy of anticancer treatment via multi-faceted effects, including immunomodulation and physiological changes. Several studies suggested that exercise effectively slows the growth of 4T1 tumor by regulating anti-tumor immunity within the body [59–62].

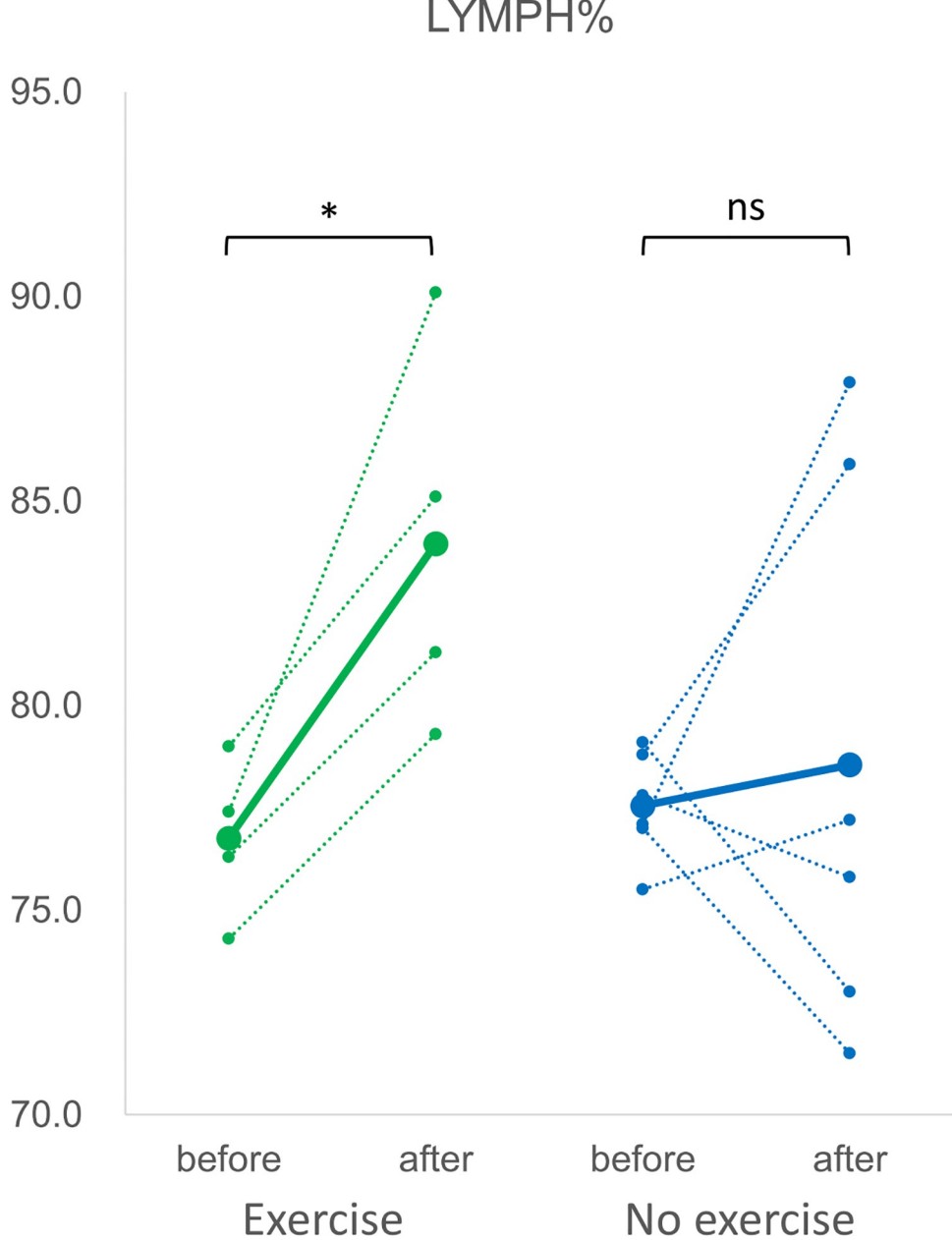

**Fig 6. The percentage of lymphocytes among white blood cells (LYMPH%).** The green dotted lines indicated the changes in LYMPH% of individual mice in the exercise group obtained before and after 6 weeks of high-intensity exercise, and the green solid line indicated their mean value. The blue dotted lines were the changes in LYMPH% of individual mice in the no exercise group before and after the duration of 6 weeks, and the blue solid line indicated their mean value. *: p<0.05. ns: not significant.

Exercise is widely recognized as a beneficial factor that reduces the incidence of cancer and improves the outcome of cancer patients from the epidemiological observations [20–24]. Nevertheless, the relation between exercise and cancer became less conclusive among preclinical studies exploring the underlying mechanisms or investigating experimental therapies with animal models [63–65]. In a systematic review collecting 53 preclinical studies about exercise

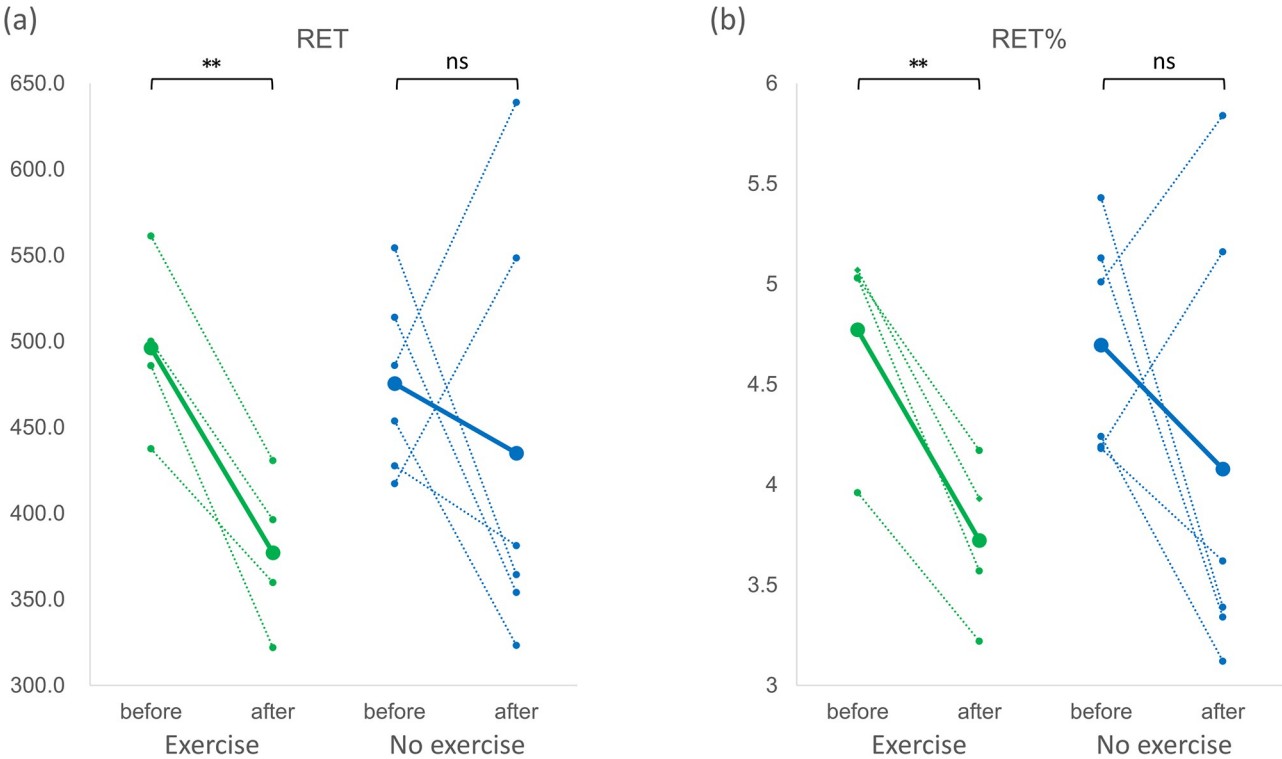

**Fig 7. The number of reticulocytes and the percentage among red blood cells.** (a) The number of reticulocytes (RET) and (b) the percentage among red blood cells (RET%). The green dotted lines indicated the changes of individual mice in the exercise group obtained before and after 6 weeks of high-intensity exercise, and the green solid line indicated their mean value. The blue dotted lines were the changes of individual mice in the no exercise group before and after the duration of 6 weeks, and the blue solid line indicated their mean value. **: $p < 0.01$. ns: not significant.

therapy in cancer, exercise reduced the incidence of cancer in 58% studies and inhibited tumor growth in 64% studies [63]. However, some studies showed no benefit or even the contrary: exercise increased cancer incidence in 8% studies and accelerated tumor growth in 9% studies [63]. In another systematic review analyzing 28 preclinical studies, similar discrepancy also existed: 5 among 20 studies reported adverse effect of exercise on tumor incidence, 2 among 14 studies reported negative impact of exercise on tumor multiplicity, and 3 among 11 studies reported increased tumor volumes by exercise [64]. The inconsistency reflects the complex interactions between exercise physiology and cancer biology, as well as great heterogeneity in experimental protocols between studies. It demands more standardized protocols including exercise treatment, disease model, and evaluation metrics to generate reproducible data and consistent conclusions, such as the Exercise as Cancer Treatment (EXACT) framework proposed by Courneya and Booth [66].

The relationship between exercise therapy and metastasis is more difficult to assess due to the limited number of preclinical studies available and the notable discrepancies observed between these studies. Sheinboim et al. found that exercise prior to tumor implantation significantly protected against metastases in distant organs in three models of melanoma [67]. The protective effects of exercise were dependent on mTOR activity and was reversed by the inhibition of the mTOR pathway with rapamycin [67]. Alvarado et al. also found in an MNU-induced model that exercised Sprague-Dawley rats developed less tumors than sedentary ones and did not develop any metastasis [68]. In addition to preventing the formation of metastasis, Vojdani et al. also showed that exercise after tumor inoculation reduced the infiltration of

metastatic 4T1 cells into liver and lung [69]. On the other hand, several studies indicated that exercise had no effect on metastasis. Yan and Demars reported that voluntary running did not reduce the number and size of pulmonary metastases as compared to sedentary control, either in experimental metastasis of B16BL/6 melanoma or spontaneous metastasis of Lewis lung carcinoma (LLC) model [70]. Similarly, Wakefield et al. concluded that exercise did not significantly affect the number of pulmonary metastasis of 4T1 cells [71]. It is noteworthy that the researchers also examined the impact of intraductal 4T1 tumor growth on exercise performance. As the tumors grew, the running behavior during the plateau phase was affected, and the running distance decreased [71]. There were even some studies claimed that exercise might have adverse effects on metastasis. MacNeil and Hoffman-Goetz reported that the tumor incidence was not different across activity groups, but the tumor multiplicity was higher in mice that had been previously exposed to exercise [72]. In a later study conducted by Smeda et al., they concluded that exercise with wheel running did not affect the volume of the primary 4T1 tumor, but the number of pulmonary metastatic nodules was significantly increased compared to the sedentary controls [73]. Standardized protocols and reporting system are required to eliminate the discrepancies between studies and reveal the true effect of exercise on metastasis.

To gain insight into the effects of exercise, we analyzed changes in blood cell components resulting from exercise. An increase in the percentage of lymphocytes following exercise was observed. Previous studies suggested that exercise induces secretion of sympathetic stimulating hormones epinephrine and norepinephrine. Sympathetic nerve stimulates the lymph system and effectively recruits lymphocytes, causing a transient increase of lymphocytes in blood [74]. The decreased neutrophil after exercise seen in our study could be explained by the observation that high-intensity exercise increases reactive oxygen species (ROS), resulting in apoptosis of neutrophils [75]. Reticulocytes are immature RBC and usually develop into mature RBC within one to three days after entering circulation. Several studies indicated that high-intensity exercise recruits reticulocytes from bone marrow and accelerates their development into mature RBC, leading to a decreased ratio of reticulocytes over all RBC [76].

Before the proposed combinatorial therapy can be translated from preclinical research to clinical practice, several issues must be addressed. Despite the rapid growing in the field of exercise oncology, only a small proportion of cancer patients participates in exercise interventions as a means to fight against cancer [77]. The barrier may result from the lack of knowledge, physical factors caused by disease or treatment, or psychological factors. Exercise interventions is generally considered safe and is recommended for cancer patients [78]. However, in some circumstances such as bone metastases or peripheral neuropathy, it requires extra caution to prescribe a safe and effective exercise program [78]. These special conditions do not preclude patients from exercise interventions, but the exercise program should be tailored by professionals who are familiar with the specific considerations of cancer and anti-cancer treatment [78]. Compliance is another problem. A considerable proportion of patients discontinued their exercise intervention programs prematurely due to various reasons such as disease progression or lack of motivation [79, 80].

There were several limitations in this study. First of all, the sample size was relatively small owing to the limited capacity of experiment instruments and operators. Increasing the number of animals may further strengthen the convincingness of the study. Second, we observed some changes in the blood cell components after exercise, but we had little information about the principal component which was responsible for the effects. Furthermore, the study only tested in one tumor model and should be generalized to other models such as cell lines originated from different organs, in different strain of mice, or even different species.

## Conclusions

In this study, we investigated the integration of exercise with pulsed-wave ultrasound hyperthermia enhanced nanochemotherapy and autophagy inhibition to treat 4T1 tumor. It was shown that mice receiving 6 weeks of high-intensity exercise before 4T1 tumor implantation had significantly smaller tumors and significantly longer survival after treatment than those received the same treatment without exercise.

## Supporting information

**S1 Data.**
(XLSX)

## Author Contributions

**Conceptualization:** Chi-Feng Chiang, Zi-Zong Wang, Yu-Hone Hsu, Shi-Chuen Miaw, Win-Li Lin.

**Formal analysis:** Chi-Feng Chiang, Zi-Zong Wang.

**Funding acquisition:** Yu-Hone Hsu, Win-Li Lin.

**Investigation:** Chi-Feng Chiang, Zi-Zong Wang.

**Methodology:** Chi-Feng Chiang, Zi-Zong Wang.

**Project administration:** Shi-Chuen Miaw, Win-Li Lin.

**Supervision:** Shi-Chuen Miaw, Win-Li Lin.

**Validation:** Zi-Zong Wang, Yu-Hone Hsu, Shi-Chuen Miaw, Win-Li Lin.

**Visualization:** Chi-Feng Chiang, Zi-Zong Wang.

**Writing – original draft:** Chi-Feng Chiang, Zi-Zong Wang.

**Writing – review & editing:** Chi-Feng Chiang, Yu-Hone Hsu, Shi-Chuen Miaw, Win-Li Lin.

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
