## [Decision Letter · Decision Letter 0]

11 Aug 2022

PONE-D-22-12073Exercise improves the outcome of anticancer treatment with ultrasound-hyperthermia-enhanced nanochemotherapy and autophagy inhibitorPLOS ONE

Dear Dr. Lin,

Thank you for submitting your manuscript to PLOS ONE. After careful consideration, we feel that it has merit but does not fully meet PLOS ONE’s publication criteria as it currently stands. Therefore, we invite you to submit a revised version of the manuscript that addresses the points raised during the review process. Can you please address the comments raised by the expert reviewer?

We look forward to receiving your revised manuscript.

Kind regards,

Avanti Dey, PhD

Staff Editor

PLOS ONE

Journal Requirements:

Reviewers' comments:

Reviewer's Responses to Questions

**Comments to the Author**

1. Is the manuscript technically sound, and do the data support the conclusions?

Reviewer #1: No

2. Has the statistical analysis been performed appropriately and rigorously? 

Reviewer #1: No

3. Have the authors made all data underlying the findings in their manuscript fully available?

Reviewer #1: No

4. Is the manuscript presented in an intelligible fashion and written in standard English?

Reviewer #1: Yes

5. Review Comments to the Author

Reviewer #1: GENERAL COMMENTS

This is a potentially interesting study but it is limited by the small sample size and the conclusions made do not reflect the data.

SPECIFIC COMMENTS

Data availability. The authors state that “relevant data are within the manuscript and its Supporting Information files”. This is not the case; data availability refers to the anonymysed raw data, not the aggregated data reported in the manuscript. In line with PLOS ONE’s data availability policy (https://journals.plos.org/plosone/s/data-availability), please make the anonymised raw data available in the supplementary materials or within a public repository.

TITLE

Please be specific in your title – exercise improved survival of anticancer treatment but it did not reduce tumour growth.

ABSTRACT

Line 40: Exercise did not reduce tumour growth. Please remove “but the difference was not significant” and make it clear in the conclusions that there was NO EVIDENCE that exercise reduced tumour growth in this experiment.

INTRODUCTION

Lines 48-49: The hallmarks of cancer were updated in 2022. Please revised accordingly. https://aacrjournals.org/cancerdiscovery/article/12/1/31/675608/Hallmarks-of-Cancer-New-DimensionsHallmarks-of

Lines 58-67: Most of the research between exercise and cancer outcomes (particularly the epidemiological evidence) relates to physical activity, not exercise per se. Exercise is a subcomponent of physical activity; it is planned, structured bouts of physical activity purposefully carried out to improve or maintain health or fitness. Please make this distinction before proceeding to discuss the impact of exercise on chemotherapeutic-related symptoms etc.

Line 68: Change “researches” to “research studies”

Line 76: When discussing the role of interleukins in the exercise-induced regulation of NK cell mobilisation and cancer cell proliferation, it is worth acknowledging this recent study which highlighted a more direct role of IL-6: https://onlinelibrary.wiley.com/doi/10.1002/ijc.33982

Line 85: What is meant by “works fine”? Please be more specific

Lines 104-109: Please mention the primary outcome(s) when discussing the aims of the study. For example, to be more specific, you might say “reduced tumour growth and recurrence” rather than “improve the outcome of treatment”.

Lines 104-109: It should be made clear in your aims that you are investigating the effect of exercise BEFORE treatment. Please amend throughout.

METHODS

Did you prospectively register the study on a publicly available database? If you did, please cite the registration. If not, please clearly state that this study was not prospectively registered.

Please provide a justification for your sample size. Here is a good paper for reference: https://online.ucpress.edu/collabra/article/8/1/33267/120491/Sample-Size-Justification

Please include a subheading that clearly details the outcomes of the study. This information seems to be under the “study design” subheading at the moment, which is not appropriate. In this section, please detail how recurrence was operationalised. Please also include a description of the methods used to analyse blood cell compartments.

Line 181: Please provide a reference to support your statement that 74.2% VO2max exceeds the threshold of high-intensity training.

Lines 175-184: Did all mice in the exercise condition complete every single planned bout of exercise?

Line 201: Tumour volume was measured every 3 days – this is a huge number of comparisons. Did you correct for multiple comparisons in your analysis? You should have done because you employed frequentist statistics, and frequentist statistics is grounded in controlling type I error rate. The results may need to be revised to incorporate error control.

RESULTS

Lines 224-226: This statement is incorrect; your study provided NO EVIDENCE that exercise strengthened the anti-tumour efficacy of treatment because there were no sig differences. Please revise accordingly,

Figures 4-6 – it is best practise to plot individual data points when you have small sample sizes (https://pubmed.ncbi.nlm.nih.gov/31608303/). Thus, please re-draw the graphs using individual data points.

DISCUSSION

Lines 284-285. Again, this statement is incorrect; your study provided NO EVIDENCE that exercise strengthened the anti-tumour efficacy of treatment because there were no sig differences. Please revise accordingly.

Please discuss the limitations of the study and the implications these limitations have on the findings. A major limitation is the small sample size.

6. PLOS authors have the option to publish the peer review history of their article (what does this mean?). If published, this will include your full peer review and any attached files.

Reviewer #1: No

---

## [Author Response · Author response to Decision Letter 0]

21 Sep 2022

Please refer to the uploaded "Response to Reviewers.docx" file for point-by-point response.

---

## [Decision Letter · Decision Letter 1]

13 Mar 2023

PONE-D-22-12073R1

Exercise improves the outcome of anticancer treatment with ultrasound-hyperthermia-enhanced nanochemotherapy and autophagy inhibitor

PLOS ONE

Dear Dr. Lin,

Thank you for submitting your manuscript to PLOS ONE. After careful consideration, we feel that it has merit but does not fully meet PLOS ONE’s publication criteria as it currently stands. Therefore, we invite you to submit a revised version of the manuscript that addresses the points raised during the review process.

We look forward to receiving your revised manuscript.

Kind regards,

Tsai-Ching Hsu, Ph.D.

Academic Editor

PLOS ONE

Additional Editor Comments:

Please revise the manuscript according to the reviewer's comments.

Reviewers' comments:

Reviewer's Responses to Questions

**Comments to the Author**

1. If the authors have adequately addressed your comments raised in a previous round of review and you feel that this manuscript is now acceptable for publication, you may indicate that here to bypass the “Comments to the Author” section, enter your conflict of interest statement in the “Confidential to Editor” section, and submit your "Accept" recommendation.

Reviewer #1: All comments have been addressed

Reviewer #2: All comments have been addressed

2. Is the manuscript technically sound, and do the data support the conclusions?

Reviewer #1: Yes

Reviewer #2: Yes

3. Has the statistical analysis been performed appropriately and rigorously? 

Reviewer #1: Yes

Reviewer #2: Yes

4. Have the authors made all data underlying the findings in their manuscript fully available?

Reviewer #1: Yes

Reviewer #2: Yes

5. Is the manuscript presented in an intelligible fashion and written in standard English?

Reviewer #1: Yes

Reviewer #2: Yes

6. Review Comments to the Author

Reviewer #1: Thank you for addressing my comments, I am satisfied with the revisions.

Reviewer #2: The manuscript is well written but needs some revision before final decision .

1) please revise the English language.

2) The 4T1 breast model is a metastatic cancer model and I extremely recommend the authors investigate the effect of the recommended treatment on metastasis of 4T1 breast cancer in vital organs.

3) The charts and figures are not prepared professionally. Please revise them in their appearance.

4) Every medical intervention has its own side effects and implications please mention yours and evaluate the side effects of your treatment in vivo.

5) Please mention the further steps for your treatment. What is its future ? which steps are needed ? What are the limitations ? etc.

7. PLOS authors have the option to publish the peer review history of their article (what does this mean?). If published, this will include your full peer review and any attached files.

Reviewer #1: **Yes: **Dr Samuel T. Orange

Reviewer #2: **Yes: **Amirhosein Kefayat

---

## [Author Response · Author response to Decision Letter 1]

27 Apr 2023

Please refer to the file named "Response to Reviewers" for the detailed responses.

---

## [Decision Letter · Decision Letter 2]

26 Jun 2023

Exercise improves the outcome of anticancer treatment with ultrasound-hyperthermia-enhanced nanochemotherapy and autophagy inhibitor

PONE-D-22-12073R2

Dear Dr. Win-Li Lin,

We’re pleased to inform you that your manuscript has been judged scientifically suitable for publication and will be formally accepted for publication once it meets all outstanding technical requirements.

Kind regards,

Tsai-Ching Hsu, Ph.D.

Academic Editor

PLOS ONE

Additional Editor Comments (optional):

All comments have been addressed.

Reviewers' comments:

Reviewer's Responses to Questions

**Comments to the Author**

1. If the authors have adequately addressed your comments raised in a previous round of review and you feel that this manuscript is now acceptable for publication, you may indicate that here to bypass the “Comments to the Author” section, enter your conflict of interest statement in the “Confidential to Editor” section, and submit your "Accept" recommendation.

Reviewer #1: All comments have been addressed

2. Is the manuscript technically sound, and do the data support the conclusions?

Reviewer #1: Yes

3. Has the statistical analysis been performed appropriately and rigorously? 

Reviewer #1: Yes

4. Have the authors made all data underlying the findings in their manuscript fully available?

Reviewer #1: Yes

5. Is the manuscript presented in an intelligible fashion and written in standard English?

Reviewer #1: Yes

6. Review Comments to the Author

Reviewer #1: I am satisfied with author responses and amendments.

7. PLOS authors have the option to publish the peer review history of their article (what does this mean?). If published, this will include your full peer review and any attached files.

Reviewer #1: **Yes: **Samuel T Orange

---

## [Editor Report · Acceptance letter]

3 Jul 2023

PONE-D-22-12073R2 

Exercise improves the outcome of anticancer treatment with ultrasound-hyperthermia-enhanced nanochemotherapy and autophagy inhibitor 

Dear Dr. Lin:

I'm pleased to inform you that your manuscript has been deemed suitable for publication in PLOS ONE. Congratulations! Your manuscript is now with our production department. 

Kind regards, 

on behalf of

Dr. Tsai-Ching Hsu 

Academic Editor

PLOS ONE